# FORMAL METHODS-BASED EVALUATION OF LLM SYSTEMATIC GENERALIZATION

## ABSTRACT

Recent advances in large language models(LLMs) have demonstrated impressive performance on a wide range of mathematical benchmarks. Yet a critical challenge remains: systematic generalization, which is the ability to correctly reason about novel combinations and unseen contexts. True systematic generalization goes beyond memorization or pattern matching, requiring models to internalize underlying rules. We present a formal methods-based evaluation framework called FORGE for rigorously probing the systematic generalization abilities of LLMs. Our approach automatically synthesizes formal benchmarks from traditional datasets, ensuring that evaluation instances are both novel and valid. Each formal benchmark is verified for correctness and well-posedness through formal methods. We further introduce a formally grounded difficulty metric and a stepwise prompting method to enhance the rigorous evaluation. Finally, we perform online evaluation multiple times and generate multiple benchmarks, ensuring novel combinations and unseen contexts for every run. Experimental results reveal a dramatic accuracy drop in top-performing LLMs, highlighting critical weaknesses of LLMs. Moreover, our analysis shows that this decline persists after controlling for problem hardness and multiple randomization, indicating that our framework not only mitigates contamination but also provides a principled scale for reasoning difficulty.

## 1 INTRODUCTION

The rapid advancement of large language models has led to impressive performance on mathematical benchmarks, from arithmetic to Olympiad-level problems (Wang et al., 2024). Despite these advances, a fundamental challenge persists: *systematic generalization* (Bahdanau et al., 2019). Systematic generalization refers to the ability of a model to correctly reason about novel combinations and unseen contexts by recomposing known elements. Unlike simple memorization or pattern matching, true systematic generalization requires models to internalize underlying rules and apply them flexibly to new situations. This ability lies at the heart of robust reasoning and compositional learning, yet it remains an open problem for current LLMs.

**Why existing evaluations fall short.** Benchmarks such as GSM8K (Cobbe et al., 2021) and MATH (Hendrycks et al., 2021) have become cornerstones for evaluating, yet their static and publicly released nature makes them highly susceptible to contamination. Prior efforts (Zhang et al., 2024; Zhu et al., 2024a; Mirzadeh et al., 2025; Fan et al., 2024; Zhu et al., 2024b) attempted to resist contamination through manually designed templates or parameter randomization. However, they introduce new challenges: templates are in limited numbers; template-driven distributions often drift in uncontrolled ways, weakening comparability; evaluations typically focus only on mathematics; natural-language prompts may admit multiple valid solutions, leaving correctness ill-defined; and difficulty is rarely quantified. A detailed comparison is shown in section 6.

Other approaches, such as (White et al., 2025), continuously update and release new datasets. Yet newly released benchmarks inevitably risk future contamination, while structural diversity across datasets hinders cross-benchmark comparability. Indeed, in 2024 alone, more than fifteen math-related datasets were published (Lookeng, 2024), further fragmenting the evaluation landscape.

**Our perspective: formalization as a remedy.** We advocate evaluating systematic generalization through formalized tasks that are synthesizable, machine-checkable, and difficulty-calibrated. We

introduce our framework **For**malized **G**eneralization **E**valuation(FORGE). Concretely, we auto-formalize seed instances from existing benchmarks into executable formal artifacts and generate fresh instances at test time via parameter randomization. Formal solvers then prove correctness or produce counterexamples, ensuring that each instance is well-posed, i.e., with guaranteed existence, uniqueness, and absence of ambiguity.

To ensure the correctness of the autoformalized benchmarks, we integrate three complementary layers of **verification**. First, static verification ensures that every formalized instance compiles as formal code, with all symbols resolving and types checking. Second, dynamic verification validates concrete parameterizations through solver execution and cross-checks across different formal methods on randomized test inputs. Finally, semantic verification employs a judge LLM to check the formal artifacts, detecting hallucinated or off-spec reasoning even when final answers appear correct.

FORGE further introduces a **difficulty metric** that characterizes problem hardness along four orthogonal dimensions: expression complexity, capturing the syntactic and structural intricacy of the problem statement; reasoning complexity, reflecting the minimal reasoning steps required to reach a solution; space and time complexity, measuring the computational effort to obtain a verified answer.

Finally, we further incorporate **stepwise prompting** by decomposing formal code into step-by-step prompts and building natural-language subquestions, thereby constructing explicit chains of thought. This design highlights the dual role of our framework: it functions as a rigorous evaluation protocol and as a potential training resource by supervising each reasoning hop with verified subgoals, which is largely absent in current LLM training pipelines.

Experimental results reveal a dramatic accuracy drop in top-performing LLMs, from approximately 95% to 60%, highlighting critical weaknesses of LLMs. Moreover, our difficulty-aware analysis shows that this decline persists even after controlling for problem hardness, indicating that our framework mitigates contamination and provides a principled scale for reasoning difficulty.

In summary, our work makes the following key contributions:

- We introduce a formal methods-based framework FORGE for evaluating LLM systematic generalization, synthesizing formal and fresh benchmarks while certifying correctness and well-posedness.
- We propose Tri-Verify (static, dynamic, semantic) to increase reliability and build stepwise prompting datasets for step-by-step training or testing.
- We develop a formally grounded difficulty metric integrating time and space complexity, expression complexity, and reasoning complexity, enabling controlled test construction and principled comparisons.
- We present experiments showing a substantial accuracy drop for strong LLMs under our settings, indicating persistent gaps in systematic generalization.

## 2 MOTIVATING EXAMPLES

To illustrate the core idea of our framework, we present two representative problems from different scientific domains.

### 2.1 N-PRIMABLE NUMBERS

A positive number is called $n$-primable if it is divisible by $n$ and all of its digits are one-digit prime numbers. The question is *"How many 3-primable positive integers are there less than 100?"*

**Formal-Natural Coupling.** We abstract the natural prompt into a parametric template: *"How many X2-primable numbers are there less than X1?"* The LLM encodes this with Z3 as:

```
valid_digits = [2, 3, 5, 7]
...
s.add(number > 0)
s.add(number < X1)
```

```
s.add(number % X2 == 0)
...
```

In cvc5, the formal code is:

```
...
(assert (> n 0))
(assert (< n X1))
(assert (= (mod n X2) 0))
...
```

For $X1 = 100, X2 = 3$, all the solvers yield exactly six valid numbers.

**Difficulty Coefficient.** We define a difficulty score $D$ by combining multiple indicators:

$$D = \alpha \cdot C_{\text{expr}} + \beta \cdot C_{\text{reason}} + \gamma \cdot C_{\text{space}} + \delta \cdot C_{\text{time}},$$

where $C_{\text{expr}}$ is the symbolic expression size, $C_{\text{reason}}$ the minimal reasoning steps, $C_{\text{space}}$ the size of the memory space, and $C_{\text{time}}$ the solver runtime. The indicators are normalized across all the benchmarks. For the original case ($X1 = 100, X2 = 3$): $C_{\text{expr}} = 3$ constraints. $C_{\text{proof}} = 6$ reasoning steps. $C_{\text{mem}} = 0.09$. $C_{\text{solve}} \approx 0.01s$. This yields a moderate difficulty score $D \approx 0.42$ (normalized).

**Controlled Randomization.** We perturb parameters within a bounded range to generate fresh but structurally equivalent tasks. For example, the query: *"How many 7-primable numbers are there less than 200?"* All the solvers find: $\{7, 35, 77\} \Rightarrow$ **3**. This randomized case ($X1 = 200, X2 = 7$) doubles the search space, raising the difficulty score to $D \approx 0.67$. However, GPT-4o incorrectly enumerates $\{7, 35, 73, 77\} \Rightarrow$ **4**, showing a reasoning slip.

**Triple Validation.** Our framework applies three layers of verification:

- **Static:** Each formal program parses and loads without errors.

- **Dynamic:** For each $(X1, X2)$, the enumerated solution sets across different formal solvers are strictly equal.

- **Semantic:** An independent judge LLM audits the formal encodings.

## 2.2 FROM A WRONG CoT TO A VERIFIED STEPWISE CHAIN

The question is *Two ice pucks collide head-on on a frictionless surface. Puck A has mass 0.691 kg and moves at 3 m/s toward stationary puck B of mass 1 kg. After the collision, puck A reverses and moves at 3 m/s opposite its original direction. Determine the impulse on puck A and on puck B.*

**LLM's Incorrect Answer.** GPT-4o produces a long chain of thoughts shown in the Appendix D but outputs a wrong final answer $\{-4.146, \ 2.073\} \, \text{N} \cdot \text{s}$, incorrectly halving puck B's impulse and violating momentum conservation at the system level.

**Formal Decomposition and Stepwise Reasoning.** Instead of one-shot generation, our formal benchmark can *factor* the solution into atomic, machine-checkable steps (units, signs, laws). Each step yields a local verdict (pass/fail) and an interpretable artifact (number, symbol, or equation). Table 1 shows the formalized code in our benchmark and the questions generated. GPT-4o will answer these questions one by one and finally output the final answer. The pipeline is: (i) extract givens; (ii) compute momenta; (iii) apply impulse definition $\Delta p$; (iv) enforce system-level conservation as a hard constraint; (v) re-compose the final answer only if all checks pass.

This example illustrates that FORGE's *formalized*, *stepwise*, and *verified* reasoning can serve two roles: (i) a stronger test (detecting globally inconsistent CoT), and (ii) training resource (supervising each reasoning hop), thereby supplying correct chains that current training pipelines lack.

Table 1: Stepwise reasoning on a 1D ice–puck collision: code, question, and answer aligned for verified chain-of-thoughts that can be used in training or testing.

| Formal Code | Generated Question | LLM's Answer |
|---|---|---|
| initialVelocityA = 3; | Q1: Initial velocity of A? | $3\,\text{m/s}$ |
| finalVelocityA = -3; | Q2: Final velocity of A? | $-3\,\text{m/s}$ |
| massA = 0.691; | Q3: Mass of A? | $0.691\,\text{kg}$ |
| massB = 1; | Q4: Mass of B? | $1\,\text{kg}$ |
| initialMomentumA = massA*initialVelocityA; | Q5: Initial momentum of A? | $2.073\,\text{kg}\cdot\text{m/s}$ |
| finalMomentumA = massA*finalVelocityA; | Q6: Final momentum of A? | $-2.073\,\text{kg}\cdot\text{m/s}$ |
| impulseA = finalMomentumA - initialMomentumA; | Q7: Impulse on A? | $-4.146\,\text{N}\cdot\text{s}$ |
| initialMomentumB = massB*0; | Q8: Initial momentum of B? | $0$ |
| finalMomentumB = initialMomentumA + initialMomentumB - finalMomentumA; | Q9: Final momentum of B? | $4.146\,\text{kg}\cdot\text{m/s}$ |
| velocityB = finalMomentumB / massB; | Q10: Velocity of B after? | $4.146\,\text{m/s}$ |
| impulseB = finalMomentumB - initialMomentumB; | Q11: Impulse on B? | $+4.146\,\text{N}\cdot\text{s}$ |
| answer = {impulseA, impulseB}; | Q12: Impulses on A and B? | $\{-4.146, +4.146\}\,\text{N}\cdot\text{s}$ |

## 3  APPROACH

We propose **FORGE**(**For**malized **G**eneralization **E**valuation), a novel benchmarking pipeline designed to transform static datasets into dynamic, verifiable, and difficulty-calibrated evaluation benchmarks. FORGE consists of six modules: (1) formal-natural coupling, (2) controlled randomization, (3) formal reasoning and verification, (4) difficulty quantification, (5) stepwise prompting, and (6) online evaluation.

**Formal-Natural Coupling**  Given a natural language problem $\mathcal{P}_{\text{NL}}$, FORGE employs an LLM to generate a corresponding formal artifact $\mathcal{P}_{\text{Code}}$. The $\mathcal{P}_{\text{Code}}$ should (i) encode the complete solution logic, (ii) maintain a one-to-one correspondence between natural language entities and formal variables, and (iii) be executable by a reasoning engine (e.g., Z3, cvc5, Mathematica). We define a binding map

$$\mathcal{B} : \mathcal{V}_{\text{NL}} \leftrightarrow \mathcal{V}_{\text{Code}},$$

where $\mathcal{V}_{\text{NL}}$ is the set of identifiable quantities in natural language and $\mathcal{V}_{\text{Code}}$ the corresponding formal variables.

**Controlled Randomization**  To mitigate contamination and test generalization, FORGE replaces constants with symbolic variables $\{x_1, x_2, \ldots, x_n\}$, sampled from constrained distributions $\mathcal{D}_{x_i}$ at test time:

$$\mathcal{P}_{\text{NL}}^{(j)}, \ \mathcal{P}_{\text{Code}}^{(j)} = \text{Perturb}(\mathcal{P}_{\text{NL}}, \mathcal{P}_{\text{Code}}, \{x_i \sim \mathcal{D}_{x_i}\}).$$

Each $\mathcal{D}_{x_i}$ respects type and constraint annotations,

$$x_i \sim \mathcal{D}_{x_i} \subseteq \mathcal{T}(x_i), \quad \text{s.t. } \phi(x_1, \ldots, x_n),$$

where $\mathcal{T}(x_i)$ is the semantic type (e.g., integer, set of digits) and $\phi$ enforces domain invariants. The randomization are controlled by the following difficulty quantification and the benchmarks are further filtered by formal verification.

**Formal Reasoning and Verification**  Given $\mathcal{P}_{\text{Code}}^{(j)}$, FORGE employs formal solvers to compute the canonical answer $y_j^*$ and enforce well-posedness including:

$$(\text{Existence}) \quad \exists y_j : \mathcal{P}_{\text{Code}}^{(j)}(x_1, \ldots, x_n) \to y_j,$$

$$(\text{Uniqueness}) \quad \forall y_1, y_2, \ \big(\mathcal{P}_{\text{Code}}^{(j)} \to y_1 \wedge \mathcal{P}_{\text{Code}}^{(j)} \to y_2\big) \ \Rightarrow \ y_1 = y_2,$$

We model verification as a composite operator

$$\mathcal{V} = \mathcal{V}_{\text{static}} \ \circ \ \mathcal{V}_{\text{dynamic}} \ \circ \ \mathcal{V}_{\text{semantic}},$$

where $\mathcal{V}_{\text{static}}$: checks well-formedness by requiring that $\Phi(b)$ compiles with all symbols resolved and types checked, $\mathcal{V}_{\text{dynamic}}$: executes solver runs on randomized parameterizations, enforcing existence and uniqueness of solutions, optionally cross-validated across independent solvers, $\mathcal{V}_{\text{semantic}}$: employs an auxiliary LLM judge to detect hallucinated or misaligned encodings even when solver outputs appear correct. An instance $f = \Phi(b)$ is admitted into the benchmark iff $\mathcal{V}(f) = \text{true}$.

**Difficulty Quantification**   Each admitted instance $f$ receives a difficulty vector
$$D(f) = \big(C_{\text{expr}}(f),\ C_{\text{reason}}(f),\ C_{\text{space}}(f),\ C_{\text{time}}(f)\big),$$
where $C_{\text{expr}}$ measures syntactic complexity, $C_{\text{reason}}$ the minimal reasoning depth, and $(C_{\text{space}}, C_{\text{time}})$ the solver resources required.

For aggregated analysis, we define a scalar difficulty score $D$ by weighting multiple indicators:
$$D = \alpha \cdot C_{\text{expr}}\ +\ \beta \cdot C_{\text{reason}}\ +\ \gamma \cdot C_{\text{space}}\ +\ \delta \cdot C_{\text{time}},$$
where $\alpha, \beta, \gamma, \delta$ are tunable coefficients reflecting the relative importance of syntactic complexity, reasoning steps, memory footprint, and runtime.

**Stepwise Prompting**   To elicit interpretable and verifiable reasoning, FORGE introduces a decomposition operator
$$\Psi : f\ \mapsto\ \langle (g_1, q_1), (g_2, q_2), \ldots, (g_k, q_k) \rangle,$$
where each pair $(g_i, q_i)$ consists of a *formal subgoal* $g_i$ derived from the structure of $f$ and a corresponding natural-language subquestion $q_i$. We extract subgoals by traversing the formal code $\mathcal{P}_{\text{Code}}$ and identifying verification checkpoints such as:

- algebraic simplifications or intermediate variable definitions,
- branching conditions in case analysis,
- inductive or iterative invariants,
- existential witnesses or constructive lemmas.

These are then ordered according to their logical dependencies, forming a directed acyclic graph that defines the sequence of subproblems. For each $g_i$, we generate a corresponding $q_i$ that verbalizes the subgoal while preserving the binding $\mathcal{B}$ between natural-language entities and formal variables.

**Online Evaluation**   We now present the new problem $\mathcal{P}_{\text{NL}}^{(j)}$ to the target LLM, which returns a prediction $\hat{y}_j$. We compare this to the canonical answer $y_j^*$ and collect multiple evaluation results:

$$\text{Exact Match:}\quad \mathbb{I}[\hat{y}_j = y_j^*]$$
$$\text{Generalization Drop:}\quad \Delta := \text{Acc}_{\text{original}} - \text{Acc}_{\text{randomized}}$$
$$\text{Multi-Level Cost:}\quad \text{Generation Cost, Reasoning Cost, Token-level Cost}$$

## 4   EXPERIMENT

To systematically evaluate FORGE, we propose the following research questions:

**RQ1: How do LLMs perform on the dynamically generated dataset?**
This question evaluates the reasoning capabilities of state-of-the-art LLMs when tested on benchmarks dynamically generated by FORGE.

**RQ2: What is the performance of LLMs with stepwise prompting?** The experiment shows the performance change of LLM with stepwise prompting generated from our benchmarks. This analysis provides insights into the feasibility of deploying FORGE to train or test LLM reasoning.

**RQ3: What is the success rate of generating formal benchmarks?**
We investigate what percentage of problems are valid, correct, well-posed, and solvable by the formal verification tool. This metric assesses the scalability of FORGE generation mechanism in producing high-quality, uncontaminated benchmarks.

**RQ4: What are the time and cost of the FORGE framework?**
We examine the computational resources required to implement FORGE, including the time needed for benchmark generation, formal verification, and model testing.

Table 2: Performance of models on GSM8K and MATH(%)

| Model | GSM8K | | | | | | MATH | | | | | |
|---|---|---|---|---|---|---|---|---|---|---|---|---|
| | Gap | Orig | R1 | R2 | R3 | Std | Gap | Orig | R1 | R2 | R3 | Std |
| Claude-3.5 | 26.75 | 98.05 | 71.55 | 70.12 | 72.22 | 1.07 | **24.46** | 85.35 | 62.27 | 61.80 | 58.61 | 1.99 |
| DeepSeek-V3 | 26.84 | 98.10 | 71.27 | 70.03 | 72.50 | 1.27 | 34.07 | 98.17 | 66.12 | 64.10 | 62.08 | 2.02 |
| GPT-4o | 31.10 | 97.44 | 66.55 | 65.62 | 66.85 | 0.64 | 28.14 | 85.16 | 58.88 | 56.78 | 55.40 | 1.75 |
| o4-mini | **17.15** | **98.66** | **82.28** | **77.04** | **82.21** | 3.01 | 29.60 | **98.71** | **71.43** | **68.13** | 67.76 | 2.02 |
| Qwen-Plus | 29.59 | 98.40 | 69.88 | 66.83 | 69.73 | 1.71 | 31.69 | 92.31 | 62.27 | 59.80 | 59.80 | 1.14 |
| Qwen2-MATH | 29.63 | 98.13 | 71.16 | 67.30 | 67.13 | 2.28 | 28.74 | 91.94 | 64.20 | 62.90 | 62.54 | 0.87 |
| QwQ | 20.61 | 98.56 | 78.28 | 76.19 | 79.38 | 1.65 | 40.83 | 97.00 | 50.75 | 52.66 | 67.10 | 7.81 |

## 4.1 EXPERIMENTAL SETTINGS

**Datasets.** We sample seed problems in the following datasets: **GSM8K** (Cobbe et al., 2021) is a dataset of grade-school math word problems requiring multi-step reasoning. **MATH** (Hendrycks et al., 2021) is a more challenging dataset covering advanced high school mathematics topics. **PHYSICS** (a Tron, 2023) is a public dataset of high school-level physics problems. **CHEM-ISTRY** (Wei et al., 2021)is a public dataset of high school-level chemistry problems.

We automatically and successfully formalize 5876 problems in the GSM8K dataset. For the other datasets, we generate 1,000 samples from each to demonstrate the generalizability of our framework across domains and problem types. Our framework is scalable, and larger formalized datasets can be constructed in future work with increased resources and funding.

**Baselines.** We tested FORGE with several state-of-the-art LLMs, including:

- GPT-4o (OpenAI) (Achiam et al., 2023): is a multilingual, multimodal generative pre-trained transformer developed by OpenAI and released in May 2024.

- o4-mini (OpenAI): is a compact, cost-efficient language model from OpenAI, optimized for fast and accurate reasoning, released in April 2025.

- Qwen-Plus-2025-04-28 (Alibaba) (Bai et al., 2023): is a large-scale open-source language model developed by Alibaba Cloud that supports both thinking and non-thinking modes.

- Qwen2-MATH (Alibaba) (Yang et al., 2024): supports using Chain-of-Thught (CoT) to solve English math problems.

- QwQ (Alibaba) (Team, 2025) is the reasoning-specialized model within the Qwen series. QwQ leverages advanced reasoning and critical thinking abilities to achieve superior performance on downstream tasks.

- Claude-3.5 Sonnet (Anthropic) (Anthropic, 2024): is an instruction-tuned LLM developed under Anthropic's constitutional AI framework and is well known for its reasoning ability.

- DeepSeek-V3 (DeepSeek) (Liu et al., 2024): We have conducted another test on DeepSeek-R1, but it takes three days for one test dataset. Due to the large deviation in the time cost, we select the DeepSeek-V3 as a more comparable baseline.

**Formal Verification Tool.** For formal verification, we employ Z3, CVC5, and Mathematica 14.0, which ensure the correctness, uniqueness, and well-posedness of generated problems.

## 4.2 RQ1: LLMs PERFORMANCE

We test each dataset three times with different randomization noted as R1, R2, and R3. The performance is summarized in Table 2and 3. The Gap metric shows the average performance decrease between the original dataset and the three dynamically generated benchmarks. The Std metric shows the standard deviation across the benchmark, indicating the stability and robustness of our framework.

Most LLMs exhibit high original accuracy but suffer the largest degradation in scientific domains, with accuracy gaps exceeding 35% in Physics and Chemistry. Qwen-MATH performs strongly in

Table 3: Performance of models on PHYSICS and CHEMISTRY(%)

| Model | PHYSICS | | | | | | CHEMISTRY | | | | | |
|---|---|---|---|---|---|---|---|---|---|---|---|---|
| | Gap | Orig | R1 | R2 | R3 | Std | Gap | Orig | R1 | R2 | R3 | Std |
| Claude-3.5 | 43.45 | **88.91** | 43.10 | 45.90 | 47.39 | 2.18 | 40.50 | **89.72** | 48.40 | 49.74 | 49.52 | 0.72 |
| DeepSeek-V3 | 23.01 | 79.71 | 60.75 | 54.52 | 54.83 | 3.51 | 27.82 | 87.49 | 61.13 | 59.12 | 58.75 | 1.28 |
| GPT-4o | 32.31 | 76.10 | 42.79 | 45.48 | 43.10 | 1.47 | 37.52 | 81.38 | 44.08 | 44.68 | 42.81 | 0.95 |
| o4-mini | **10.15** | 79.56 | **68.46** | **68.94** | 70.84 | 1.26 | **4.39** | 80.19 | **75.20** | **76.77** | 75.43 | 0.85 |
| Qwen-Plus | 46.21 | 88.27 | 42.47 | 44.10 | 39.60 | 2.28 | 35.69 | 79.97 | 44.90 | 43.93 | 44.01 | 0.54 |
| Qwen2-MATH | 27.81 | 69.09 | 41.37 | 42.47 | 40.01 | 1.23 | 22.83 | 66.71 | 45.27 | 42.81 | 43.56 | 1.26 |
| QwQ | 18.59 | 79.23 | 61.72 | 59.77 | 60.42 | 1.02 | 8.70 | 83.69 | 74.70 | 75.15 | 74.11 | 0.24 |

Table 4: Performance (%) comparison of single-shot and stepwise prompting across four domains.

| | GSM8K | | MATH | | PHYSICS | | CHEMISTRY | |
|---|---|---|---|---|---|---|---|---|
| | Single-shot | Step-wise | Single-shot | Step-wise | Single-shot | Step-wise | Single-shot | Step-wise |
| R1 | 82.28 | 84.23 | 71.43 | 70.23 | 68.46 | 63.19 | 75.20 | 72.61 |
| R2 | 77.04 | 79.86 | 68.13 | 67.71 | 68.94 | 66.28 | 76.77 | 72.76 |
| R3 | 82.21 | 84.25 | 67.76 | 69.54 | 70.84 | 67.42 | 75.43 | 72.69 |

mathematics but generalizes poorly to physics and chemistry, revealing domain-specific limitations. QwQ shows a large decrease (40%) on MATH datasets but a relatively small decrease on the other three datasets. By contrast, o4-mini demonstrates superior robustness and retains high accuracy on all the datasets after randomization. It validates the strong performance of o4-mini in multiple domains and benchmarks. Appendix E further shows the difficulty metrics across different domains.

Detailed error analysis revealed that LLMs struggled with algebraic reasoning, especially when symbolic structure conflicted with learned patterns. In particular, some models incorrectly treated linear expressions as quadratics, leading to invalid root-based answers. Additionally, misinterpretation of randomized numerical variables was a common failure mode. For instance, values representing proportions were sometimes mistaken for absolute quantities, producing incorrect equations. These issues highlight reasoning gaps in symbolic parsing and context-dependent numerical understanding.

### 4.3 RQ2: STEPWISE PROMPTING

We next evaluate the impact of stepwise prompting on o4-mini, where formal code is decomposed into intermediate reasoning steps and translated into natural language sub-questions. As shown in Table 4, stepwise prompting shows slightly improved performance on GSM8K. However, its effect is mixed in scientific domains: accuracy drops in PHYSICS and CHEMISTRY. Decomposition lengthens the output; with per-step error rate $\epsilon$, the chance the *entire* chain is correct shrinks roughly as $(1 - \epsilon)^T$ with more steps $T$, unless strong verification and consistency checks are present. Besides, LLM may not be explicitly trained to follow strict step-by-step reasoning. Without step-supervised training, models may treat steps as stylistic rather than semantic constraints. The results show that stepwise prompting is a useful *probe* for analyzing reasoning behavior.

### 4.4 RQ3: SUCCESS RATE OF AUTOMATIC FORMAL BENCHMARK GENERATION

This experiment measures how reliably FORGE can generate valid and well-posed benchmarks. For each domain, we conduct tri-verify to check whether they (i) compile into executable formal code, (ii) dynamically execute the formal code and verify with the ground truth and cross-check across different formal methods if they exist, and (iii) employ a judge LLM to confirm the semantics. Results in Table 5 report the success rate as the fraction of problems that pass three verification. Overall, the tri-verification pipeline ensures that only high-quality problems are retained, maintaining both syntactic validity and semantic faithfulness.

### 4.5 RQ4: COMPUTATIONAL COSTS

We measure the time and token costs of implementing FORGE, covering three main stages: (i) code generation, where LLMs produce formalized code; (ii) problem randomization and formal

Table 5: Success rate (%) of automatic formal benchmark generation after tri-verification.

| Domain | Static Check | Dynamic Check | Semantic Check |
|---|---|---|---|
| GSM8K | 97.62 | 91.42 | 96.29 |
| MATH | 85.01 | 89.78 | 91.30 |
| Physics | 91.23 | 83.17 | 97.30 |
| Chemistry | 84.47 | 81.54 | 95.97 |

Table 6: Per-sample token cost and time cost of LLMs on four datasets.

| Model | GSM8K | | | | MATH | | | |
|---|---|---|---|---|---|---|---|---|
| | R1 | R2 | R3 | Time | R1 | R2 | R3 | Time |
| Claude-3.5 | 132.94 | 133.39 | 130.76 | 4.40s | 239.13 | 238.55 | 231.65 | 4.83s |
| DeepSeek-V3 | 176.82 | 177.28 | 180.27 | 10.50s | 534.95 | 541.54 | 521.38 | 20.30s |
| GPT-4o | 202.11 | 203.23 | 206.16 | 3.48s | 404.35 | 400.33 | 408.30 | 6.25s |
| o4-mini | 25.05 | 23.65 | 23.53 | 5.45s | 161.14 | 30.09 | 29.60 | 12.11s |
| Qwen-Plus | 184.23 | 174.78 | 159.14 | 11.42s | 503.00 | 490.30 | 505.80 | 26.67s |
| Qwen2-Math | 172.42 | 173.62 | 173.60 | 7.32s | 526.31 | 525.26 | 524.67 | 13.21s |
| QwQ | 44.41 | 46.43 | 49.48 | 18.85s | 277.27 | 238.70 | 244.82 | 30.85s |

| Model | PHYSICS | | | | CHEMISTRY | | | |
|---|---|---|---|---|---|---|---|---|
| | R1 | R2 | R3 | Time | R1 | R2 | R3 | Time |
| Claude-3.5 | 191.50 | 190.63 | 191.10 | 3.52s | 167.99 | 168.01 | 168.70 | 4.13s |
| DeepSeek-V3 | 325.33 | 321.12 | 329.19 | 12.55s | 269.91 | 269.49 | 268.68 | 13.10s |
| GPT-4o | 347.50 | 319.27 | 340.39 | 5.85s | 296.50 | 294.40 | 296.30 | 6.50s |
| o4-mini | 129.31 | 38.42 | 34.79 | 15.88s | 32.15 | 31.99 | 31.60 | 15.15s |
| Qwen-Plus | 388.10 | 380.70 | 389.70 | 18.29s | 294.10 | 291.40 | 290.30 | 17.14s |
| Qwen2-Math | 426.74 | 424.98 | 418.45 | 10.88s | 316.85 | 317.59 | 317.55 | 15.30s |
| QwQ | 253.44 | 223.08 | 234.17 | 16.81s | 198.76 | 181.07 | 203.38 | 16.32s |

verification, where formal tools solve or validate problems under varied settings; and (iii) model testing, where LLMs are evaluated on three randomized datasets.

Token per instance of formal code remains modest across tasks, averaging around 55.69 tokens for GSM8K and 52.72 MATH, and slightly higher for Physics (64.82) and Chemistry (65.90). The cost of tokens of LLMs is listed in Table 6. During online testing, we observed noticeable variation in token consumption across models and domains. On GSM8K, token usage was relatively low and consistent, while tasks like MATH, Physics, and Chemistry exhibited significantly higher usage, particularly for models like ChatGPT-4o and Qwen-Plus. While MATH problems are symbolically simpler to formalize, they incur significantly higher LLM inference cost in both token consumption and latency. Such findings further motivate the need for symbolic supervision in benchmarking and the use of formally verified outputs as ground truth. Finally, randomization and formal verification added about 0.9 seconds per problem. It indicates that although FORGE introduces additional computational demands, its scalability and benefits outweigh the costs.

## 5 DISCUSSION

**Beyond correctness: broader benefits of Formalization**  Formalization not only mitigates contamination but also brings broader benefits for evaluation and pedagogy. It enforces *rigor*, as machine-checked proofs prevent hand-waving and expose ambiguity early. It enhances *transparency*, since proof scripts are replayable, intermediate states are inspectable, and counterexamples surface incorrect claims. Moreover, difficulty becomes *quantifiable*: proof length, state-space size, and solver runtime provide principled metrics beyond heuristic labels. Finally, formal structures naturally support *taxonomy and well-posedness*, grouping problems by type and filtering degenerate cases, while parameterized generation yields fresh uncontaminated instances for every run.

**Limitations**  While FORGE shows great significance, several limitations remain. First, solver-based verification introduces a modest computational overhead (approximately 0.1–1s per problem), though

Table 7: Comparison of recent related evaluation methods

| Title | Mutation | Validation | Templates Amount |
|---|---|---|---|
| MPA | LLM/Human | LLM/Human | about 15000 |
| DYVAL | Predefined rules | DAG Algorithm | - |
| GSM-Symbolic | Predefined rules | Human and Algorithm | 50 |
| NPHardEval | Monthly update | Algorithm | 900 |
| LIVEBENCH | Monthly update | Automated scoring tools | about 1000 |
| FORGE (ours) | Difficulty-aware | Formal Verification | about 9000 |

this cost is negligible compared to the typical inference latency of modern LLMs. Second, our current implementation leverages only three formal tools, which limits the coverage of certain domains, particularly those requiring open-ended or higher-order reasoning. Expanding support for richer formal systems is ongoing work. Furthermore, due to time and budget constraints, we conduct experiments on about 9k test instances, and our evaluation focuses on five widely used LLMs. The overall cost is about 3673 USD. We are committed to scaling both the dataset and model coverage in future iterations, contingent on additional resources and funding.

## 6 RELATED WORK

**Dataset Generation**    Recently, several dynamic or contamination-limited benchmarks have been introduced. Meta Probing Agents (MPA) (Zhu et al., 2024b) reformulate benchmark questions by paraphrasing and inserting distractors, with verification performed by judge agents and human annotators. DYVAL (Zhu et al., 2024a) generates reasoning tasks through Directed Acyclic Graph(DAG) structures, adjusting difficulty by controlling depth and width, with about 500 instances per task. GSM-Symbolic (Mirzadeh et al., 2025) constructs 100 symbolic templates from GSM8K, each expanded into 50 variants, diversified by name and number substitutions and clause modifications. NPHardEval (Fan et al., 2024) builds 900 problems across three complexity classes and verifies them using established algorithms. LiveBench (White et al., 2025) focuses on contamination-resistance, sourcing problems from newly released competitions, news, and datasets, refreshing one-sixth of its tasks monthly. Table 7 shows a summary of these related methods. Unlike these earlier pipelines, our method ensures structural and semantic rigor, promoting fair and reproducible assessment. Beyond mathematics, our framework is general and has been extended to physics and chemistry.

**Mathematics Benchmarks**    Many datasets have been proposed to evaluate LLMs' mathematical reasoning abilities, such as MATH (Hendrycks et al., 2021) and GSM8K (Cobbe et al., 2021). As model accuracy has risen, e.g., 87.9% on MATH (Lei et al., 2024) and 97.1% on GSM8K (Zhong et al., 2025), these benchmarks are becoming saturated and less effective at differentiating models. Newer datasets like ARB (Sawada et al., 2023), OlympiadBench (He et al., 2024), and SciBench (Wang et al., 2024) aim to increase difficulty, but often include tasks that require human evaluation. Attempts to automate this using rubric-guided LLMs have proven unreliable. Putnam-AXIOM (Gulati et al., 2024) offers a curated benchmark with standardized boxed answers for automatic evaluation.

**Data Contamination**    Contamination, where evaluation data appears in pretraining, has been highlighted by Brown et al. (Brown et al., 2020) and further analyzed by Carlini et al. (Carlini et al., 2023) and Kandpal et al. (Kandpal et al., 2022). Razeghi et al. (Razeghi et al., 2022) showed performance improves with exposure frequency in pretraining corpora. Unlike these retrospective studies, our method avoids contamination by generating fresh problems with formal verification.

## 7 CONCLUSION

In this work, we introduced a formal method-based framework FORGE for evaluating the systematic generalization of LLMs. By synthesizing fresh and formal benchmarks and validating them through multi-level verification, our approach ensures that evaluations remain robust and reflective of genuine generalization ability. Beyond mathematics, the framework is general and can be extended to other domains, opening the door to richer evaluations and broader scientific impact.

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

## A  REPRODUCIBILITY STATEMENT

We publish our framework and the dataset on the anonymous website(anonymous, 2025).

## B  ETHICS STATEMENT

Our work aims to improve the reliability and transparency of LLM evaluation through formal verification. The datasets used in this paper are either publicly available benchmarks (e.g., GSM8K, MATH, Physics) or are automatically generated through controlled randomization and formal verification, without involving any personally identifiable or sensitive information. The research does not pose foreseeable risks to individuals or groups. Potential misuse, such as applying our benchmark generation pipeline to create adversarial test sets for misleading model evaluation, should be mitigated by releasing both the methodology and verification artifacts for reproducibility and transparency. We encourage responsible use of our framework in scientific and educational contexts.

## C  USE OF LLMS

We employed LLMs as components of our pipeline, specifically for translating natural language problems into formal code. The models used such as OpenAI's GPT and Alibaba's Qwen, accessed through publicly available APIs. LLMs were not used to write the main body of this paper; instead, they were limited to experimental components described above. All outputs generated by LLMs were subject to human verification or formal reasoning checks before inclusion in our study.

## D  MOTIVATING EXAMPLE

The origin output of GPT-4o is

> To find the impulse on each puck, we'll use the principles of conservation of momentum and the definition of impulse.
>
> **1. Conservation of Momentum:**
>
> Before the collision:
>
> - The momentum of puck A $= m_A \times v_A = 0.691\,\text{kg} \times 3\,\text{m/s} = 2.073\,\text{kg m/s}$
> - The momentum of puck B $= 0\,\text{kg m/s}$ (since it is stationary).
>
> After the collision:
>
> - The momentum of puck A $= m_A \times (-v_A) = 0.691\,\text{kg} \times (-3)\,\text{m/s} = -2.073\,\text{kg m/s}$
>
> **2. Impulse Calculation:**
>
> The impulse on an object is the change in momentum of the object.
>
> Impulse on puck A:
>
> - Change in momentum of puck A = final momentum $-$ initial momentum $= -2.073\,\text{kg m/s} - 2.073\,\text{kg m/s}$
> - Change in momentum for puck A $= -4.146\,\text{kg m/s}$
>
> Therefore, the impulse on puck A is $-4.146\,\text{kg m/s}$.
>
> **3. Impulse on puck B:**
>
> By conservation of momentum, the total initial momentum ($2.073\,\text{kg m/s}$ for the system) is equal to the total final momentum of the system.
>
> The total momentum after the collision should be zero because puck A's momentum is $-2.073\,\text{kg m/s}$.
>
> Therefore, puck B's momentum would be $2.073\,\text{kg m/s}$ to balance the negative momentum of puck A.
>
> Since puck B started with zero momentum and ended with $2.073\,\text{kg m/s}$, the impulse experienced by puck B is:

Table 8: Difficulty metrics (avg/std) across domains.

| | GSM | | | | MATH | | | |
| | time | space | expr | reason | time | space | expr | reason |
| --- | --- | --- | --- | --- | --- | --- | --- | --- |
| origin | 0.12/0.12 | 0.11/0.13 | 0.17/0.14 | 0.15/0.10 | 0.16/0.05 | 0.09/0.09 | 0.16/0.09 | 0.18/0.12 |
| random_1 | 0.58/0.06 | 0.12/0.15 | 0.16/0.14 | 0.15/0.10 | 0.27/0.14 | 0.03/0.10 | 0.15/0.18 | 0.14/0.19 |
| random_2 | 0.47/0.05 | 0.15/0.18 | 0.13/0.10 | 0.09/0.05 | 0.26/0.04 | 0.03/0.09 | 0.15/0.09 | 0.17/0.10 |
| random_3 | 0.59/0.07 | 0.12/0.13 | 0.11/0.09 | 0.11/0.07 | 0.28/0.13 | 0.13/0.13 | 0.14/0.18 | 0.14/0.19 |
| | Physics | | | | Chemistry | | | |
| | time | space | expr | reason | time | space | expr | reason |
| origin | 0.01/0.05 | 0.04/0.07 | 0.21/0.14 | 0.23/0.14 | 0.06/0.20 | 0.14/0.20 | 0.14/0.16 | 0.22/0.18 |
| random_1 | 0.06/0.16 | 0.32/0.15 | 0.16/0.23 | 0.19/0.21 | 0.05/0.19 | 0.14/0.19 | 0.13/0.16 | 0.21/0.18 |
| random_2 | 0.06/0.15 | 0.22/0.08 | 0.15/0.24 | 0.17/0.22 | 0.05/0.19 | 0.06/0.21 | 0.15/0.15 | 0.17/0.17 |
| random_3 | 0.07/0.16 | 0.26/0.12 | 0.16/0.23 | 0.19/0.21 | 0.04/0.20 | 0.06/0.20 | 0.14/0.15 | 0.21/0.18 |

- $\Delta p = 2.073 \, \text{kg m/s}$

The impulses are equal in magnitude but opposite in direction, which aligns with Newton's third law.

The answer is: $-4.146, 2.073$.

## E    DIFFICULTY CALIBRATION

We show the difficulty analysis of all the generated benchmarks tested in the experiment section in Table 8. Across all four domains, different randomization within the same dataset yield similar difficulty metrics, while showing similar values compared to the original problems.

## F    CASE STUDIES

**Example 1**    *Find the 2 distinct solutions to the equation $x^2 - 6x - 60 = -5$.*

**Ground Truth:** $x_1 = -5$, $x_2 = 11$

**LLM Output:** The answer is: $x_1 = -5$, $x_2 = 11$

**Formalized Code:**

```
solutions = Solve[x^2 - 6 x - 60 == -5, x];
answer = x /. solutions;
```

**Randomized Version:** *Find the 1 distinct solution to the equation $x^1 - 7x - 54 = -3$.*

**Ground Truth:** $-\frac{17}{2}$

**LLM Output:** The answer to this equation is $\frac{7+\sqrt{253}}{2}$ and $\frac{7-\sqrt{253}}{2}$

**Formalized Code:**

```
solutions = Solve[x^1 - 7 x - 54 == -3, x];
answer = x /. solutions;
```

**Example 2**    *Mathilda is determined to pay back the money she owes a friend, so she decides to pay an initial installment of \$60. If she still has 0.4 left to pay, how much did she owe originally?*

**Ground Truth:** 100

**LLM Output:** Assuming that the money she owed is $x$,The solution is $x - 60 = 0.4x$,which means the answer is \$100

**Formalized Code:**

```
(* Let x be the total amount Mathilda owed *)
eqn = x - 60 == 0.4 x;
solution = Solve[eqn, x];
answer = x /. solution[[1]]
```

**Randomized Version:** *Mathilda is determined to pay back the money she owes a friend, so she decides to pay an initial installment of \$72. If she still has 0.537 left to pay, how much did she owe originally?*

**Ground Truth:** 155.51

**LLM Output:** The answer is: $x = 72 + 0.537 = 72.537$

**Formalized Code:**

```
(* Let x be the total amount Mathilda owed *)
eqn = x - 72 == 0.537 x;
solution = Solve[eqn, x];
answer = x /. solution[[1]]
```

**Example 3** *A proposal will make years that end in double zeroes a leap year only if the year leaves a remainder of 200 or 600 when divided by 900. Under this proposal, how many leap years will there be that end in double zeroes between 1996 and 4096?*

**Ground Truth:** 5

**LLM Output:** The answer is: 5

**Formalized Code:**

```
years = Range[2000, 4096, 100];
leapYears = Select[years, Mod[#, 900] == 200 || Mod[#, 900] == 600 &];
answer = Length[leapYears];
answer
```

**Randomized Version:** *A proposal will make years that end in double zeroes a leap year only if the year leaves a remainder of 286 or 469 when divided by 494. Under this proposal, how many leap years will there be that end in double zeroes between 1996 and 5750?*

**Ground Truth:** 0

**LLM Output:** To solve the problem, we need to determine how many years ending in double zeroes between 1996 and 5750 satisfy the condition of leaving a remainder of 286 or 469 when divided by 494. 1. Determine the range of years ending in double zeroes from 1996 to 5750. These years are: 2000, 2100, 2200, ..., 5700. 2. Verify if these years are divisible by 100. Since we're only considering years ending in double zeroes, this step is satisfied. 3. For each year $y$ in the list, calculate $y \mod 494$. We need:

$$y \equiv 286 \pmod{494} \quad \text{or} \quad y \equiv 469 \pmod{494}$$

4. The sequence of years ending in double zeroes starting from 2000 is an arithmetic sequence defined as $2000 + 100k$ where $k$ is a non-negative integer. 5. Calculate the values of $k$ such

that the above congruence conditions are satisfied: - $2000 + 100k \equiv 286 \pmod{494}$ - Simplify: $100k \equiv 286 - 2000 \equiv 286 \equiv -240 \pmod{494}$ - Solve for $k$: $100k \equiv 254 \pmod{494}$

- The inverse of 100 modulo 494 can be determined using the Extended Euclidean Algorithm: - $494 = 100 \times 4 + 94$ - $100 = 94 \times 1 + 6$ - $94 = 6 \times 15 + 4$ - $6 = 4 \times 1 + 2$ - $4 = 2 \times 2 + 0$ - $2 = 2 \times 1$ - Work backwards to find coefficients - Inverse is found to be 198

- Therefore, multiplying both sides of $100k \equiv 254 \pmod{494}$ by 198 gives: - $k \equiv 254 \times 198 \equiv 4 \pmod{494}$ 6. Now form concrete values of years: - $2000 + 100 \times 4 = 2400$ - Continue adding values for satisfying condition $k = 4$: - Using a similar method, find values for second condition.

Create list:

- Evaluate for years: 2400, 2894, 3388, ..., 5626
- Length of sequence: Using the arithmetic sequence formula, find total number years satisfying each congruence.

Count these years for condition $k \equiv 240$.

The answer is: 6

**Formalized Code:**

```
years = Range[2000, 5750, 100];
leapYears = Select[years, Mod[#, 494] == 286 || Mod[#, 494] == 469 &];
answer = Length[leapYears];
answer
```

**Example 4** *Let $m$ be the smallest positive, three-digit integer congruent to 5 (mod 11). Let $n$ be the smallest positive, four-digit integer congruent to 5 (mod 11). What is $n - m$?*

**Ground Truth:** 902

**LLM Output:** Let me solve this step by step:

1) For $m$, we need: * The smallest three-digit number (100) to find where to start * Numbers congruent to 5 (mod 11) have the form 11k + 5, where k is a non-negative integer

2) Let's find $m$: * Starting with k = 9: 11(9) + 5 = 104 (this is our first three-digit number) * Therefore, $m = 104$

3) For $n$, we need: * The smallest four-digit number (1000) to find where to start * Same form: 11k + 5

4) Let's find $n$: * We need the first value of 11k + 5 ≥ 1000 * k = 90: 11(90) + 5 = 995 (too small) * k = 91: 11(91) + 5 = 1006 (this is our number) * Therefore, $n = 1006$

5) Calculate $n - m$: * 1006 - 104 = 902

The answer is: 902

**Formalized Code:**

```
m = Min[Select[Table[i, {i, 100, 199}], Mod[#, 11] == 5 &]];
n = Min[Select[Table[i, {i, 1000, 1999}], Mod[#, 11] == 5 &]];
answer = n - m;
answer
```

**Randomized Version:** *Let $m$ be the smallest positive, three-digit integer congruent to 2 (mod 12). Let $n$ be the smallest positive, four-digit integer congruent to 2 (mod 12). What is $n - m$?*

**Ground Truth:** 900

**LLM Output:** To solve the problem, we need to find two specific integers, $m$ and $n$, based on their congruence properties and then calculate $n - m$.

1. **Finding $m$:**

We want the smallest positive three-digit integer $m$ such that $m \equiv 2 \pmod{12}$.

Start with the smallest three-digit number, which is 100.

Check $100 \mod 12$:

$$100 \div 12 = 8 \quad \text{remainder: } 100 - 8 \times 12 = 100 - 96 = 4$$

So, $100 \equiv 4 \pmod{12}$.

To get $m \equiv 2$, we need:

$$m = 100 - 4 + 2 = 98 \quad \text{(but this is not a three-digit number)}$$

Let's try adding multiples of 12 to 100 until we achieve a congruence of 2 modulo 12:

$$104 = 100 + 4, \quad 104 \equiv 8 \pmod{12}$$
$$106 = 104 + 2, \quad 106 \equiv 10 \pmod{12}$$
$$108 = 106 + 2, \quad 108 \equiv 0 \pmod{12}$$
$$110 = 108 + 2, \quad 110 \equiv 2 \pmod{12}$$

Thus, $m = 110$.

2. **Finding $n$:**

We want the smallest positive four-digit integer $n$ such that $n \equiv 2 \pmod{12}$.

Start with the smallest four-digit number, which is 1000.

Check $1000 \mod 12$:

$$1000 \div 12 = 83 \quad \text{remainder: } 1000 - 83 \times 12 = 1000 - 996 = 4$$

So, $1000 \equiv 4 \pmod{12}$.

To get $n \equiv 2$, we need:

$$n = 1000 - 4 + 2 = 998 \quad \text{(but this is not a four-digit number)}$$

Let's try adding multiples of 12 to 1000 until we achieve a congruence of 2 modulo 12:

$$1004 = 1000 + 4, \quad 1004 \equiv 8 \pmod{12}$$
$$1006 = 1004 + 2, \quad 1006 \equiv 2 \pmod{12}$$

Thus, $n = 1006$.

3. **Calculating $n - m$:**

We find:

$$n - m = 1006 - 110 = 894$$

**Formalized Code:**

```
m = Min[Select[Table[i, {i, 100, 199}], Mod[#, 12] == 2 &]];
n = Min[Select[Table[i, {i, 1000, 1999}], Mod[#, 12] == 2 &]];
answer = n - m;
answer
```

**Example 5** *When simplified, what is the value of $\sqrt{3} \times 3^{\frac{1}{2}} + 12 \div 3 \times 2 - 4^{\frac{3}{2}}$?*

**Ground Truth:** 3

**LLM Output:** Let me solve this step by step:

1) First, let's simplify $\sqrt{3} \times 3^{\frac{1}{2}}$ * Note that $3^{\frac{1}{2}} = \sqrt{3}$ * So this becomes $\sqrt{3} \times \sqrt{3} = 3$

2) Next, let's simplify $4^{\frac{3}{2}}$ * $4^{\frac{3}{2}} = (2^2)^{\frac{3}{2}} = 2^3 = 8$

3) For $12 \div 3 \times 2$ * Going from left to right: $12 \div 3 = 4$ * Then $4 \times 2 = 8$

4) Now we have: * $3 + 8 - 8$

5) Final calculation: * $3 + 8 - 8 = 3$

The answer is: 3

**Formalized Code:**

```
answer = N[Sqrt[3] * 3^(1/2) + 12 / 3 * 2 - 4^(3/2)]
```

**Randomized Version:** *When simplified, what is the value of $\sqrt{8} \times 8^{\frac{2}{7}} + 70 \div 8 \times 7 - 63^{\frac{8}{7}}$?*

**Ground Truth:** $-47.4912$

**LLM Output:** ach part of the expression step by step:

1. Simplify $\sqrt{8} \times 8^{\frac{2}{7}}$: - $\sqrt{8} = 8^{\frac{1}{2}}$. - So, we have $8^{\frac{1}{2}} \times 8^{\frac{2}{7}}$. - When multiplying powers with the same base, we add the exponents:

$$8^{\frac{1}{2} + \frac{2}{7}}.$$

- Find a common denominator for $\frac{1}{2}$ and $\frac{2}{7}$, which is 14:

$$\frac{1}{2} = \frac{7}{14}, \quad \frac{2}{7} = \frac{4}{14}.$$

- Add the exponents:

$$\frac{7}{14} + \frac{4}{14} = \frac{11}{14}.$$

- Thus, $8^{\frac{11}{14}}$.

2. Evaluate $70 \div 8 \times 7$: - First, divide 70 by 8:

$$\frac{70}{8} = \frac{35}{4}.$$

- Then, multiply by 7:

$$\frac{35}{4} \times 7 = \frac{245}{4}.$$

3. Simplify $63^{\frac{8}{7}}$: - Rewrite $63^{\frac{8}{7}}$ as $(63^{\frac{1}{7}})^8$. - Notice that $63 = 7^2 \times 3$. - So, $63^{\frac{1}{7}} = (7^2 \times 3)^{\frac{1}{7}} = 7^{\frac{2}{7}} \times 3^{\frac{1}{7}}$. - Evaluate $(7^{\frac{2}{7}} \times 3^{\frac{1}{7}})^8$:

$$(7^{\frac{2}{7}})^8 \times (3^{\frac{1}{7}})^8 = 7^{\frac{16}{7}} \times 3^{\frac{8}{7}}.$$

To pause here on simplification, since it's potentially reducible further but conforms to our problem scope, we will execute the main expression:

Bring together:

Then, expression becomes:

Move to completion on expression: Expression results before further combination, primary constants handled as $245, 7^x, 3^{x'}$)

Combining we have $8^{\frac{11}{14}} + \frac{245}{4} - 63^{\frac{8}{7}}$.

Therefore, we stop simplification, representation of exact values:

combined assessment yields: The answer is: $8^{\frac{11}{14}} + \frac{245}{4} - 63^{\frac{8}{7}}$ which is 0.

Finally establish: The answer is: 0

**Formalized Code:**

```
answer = N[Sqrt[8]*8^(2/7) + 70/8*7 - 63^(8/7)]
```

