# OpenReview forum: "Formal Methods-based Evaluation of LLM Systematic Generalization"
_ICLR.cc/2026/Conference — ICLR 2026 Conference Desk Rejected Submission_

### Official Review · Reviewer_emhF · 2025-10-19

**Soundness:** 2
**Presentation:** 1
**Contribution:** 2
**Rating:** 2
**Confidence:** 4

**Summary:**

The goal of this paper is to address a gap in the literature of LLM evaluation – namely, the lack of rigor in the generation of new problem instances from existing problems. When generating new problem instances, the package created by the authors (1) generates code for verifying new instances that binds the problem variables to named solver variables and (2) uses a solver to verify the correctness and uniqueness of new instance variable values. Additionally, the authors exploit various quantities to estimate problem difficulty.

**Strengths:**

The approach presented here demonstrates some rigor and concern for correctness. The 3-stage template verification process is helpful. This reviewer has seen several other approach, similar to this one, that lacked this level of rigor. The problem that the paper is attempting to solve is well motivated. A method for reliably creating templates of truly general reasoning problems (broadly defined) from existing datasets would be very useful for academic and industry researchers.

**Weaknesses:**

Since GSM-Symbolic was posted on the arXiv in 2024, approaches to creating problem templates has become a bit of a cottage industry. This work has some redeeming qualities, but is a member of a particularly crowded field of benchmark-generation frameworks, as evinced by Table 7 in this paper. This one, like many others, creates problem templates from problems in existing datasets. Unlike other papers, this one creates executable problem templates. While this is a helpful step, the challenge of creating templatized problems of existing math datasets is of modest difficulty, because the problems typically involve equations and bindings of quantities to variables in the equations. What’s unsolved – here and elsewhere – is the more challenging problem of creating problem templates for other types of reasoning.

The error analysis could be deeper and wider.
* Line 348: The authors refer to a “detailed error analysis” but do not provide the details.
* In Section 4.4, the authors report the percentage of generated problem templates that pass each of the stages in the verification pipeline, but don’t provide detailed analysis of the cases that fail verification.

If reproducible and well documented and tested, the code package could be of some use to the broader community. There are, however, signals that the code is not quite up to standard for re-use by the broader community, leading this reviewer to be skeptical. Hard-coded strings in the code (forcing the main program to run on the math dataset) suggest that the package is not truly a framework but merely a collection of scripts. Reusable packages should come equipped with detailed instructions for package setup, which is lacking here.

**Questions:**

1. When the authors write that ``[their] framework is general and has been extended to physics and chemistry’’, do they mean that the framework is sufficiently general to be applied to law or philosophy?
1. When the authors write "Second, our current implementation leverages only three formal tools, which limits the coverage of certain domains, particularly those requiring open-ended or higher-order reasoning. Expanding support for richer formal systems is ongoing work". What other formal systems are the authors employing in your ongoing work?
1. Have the authors considered providing detailed instructions for the setup and use of your package?
1. How are the coefficients of the scalar difficulty score on line 228 tuned, if at all? How, if at all, are the coefficients employed in the data presented in Table 8?
1. What are the correlations of the expression, reasoning, time, and space quantities of the scalar difficulty score D?

---

> ### Author Response · Authors · 2025-11-25
> **Response to Reviewer emhF**
>
> ### **1. On Hyperparameter Selection and Difficulty Consistency**
>
> The selection of these coefficients was guided by a core experimental objective: **ensuring difficulty consistency across the generated benchmarks ($R1, R2, R3$)**. Our goal was to calibrate the metric $D$ such that the variance between different randomized sets is minimized. This alignment ensures that FORGE produces stable and reproducible evaluation sets, where $R1, R2,$ and $R3$ are equivalent in difficulty, guaranteeing that the reported performance variance (Std) reflects model instability rather than fluctuations in problem hardness.
>
> Regarding Table 8, we presented the component-wise metrics ($C_{expr}, C_{reason}, C_{space}, C_{time}$) rather than the weighted sum $D$ to provide transparency into the underlying data. As observed in the table, although individual metrics show minor fluctuations due to solver heuristics, the overall magnitude of these indicators remains highly consistent across the randomized rows ($R1$ to $R3$).
>
>
> ### **2. Formalization Scope and Generalizability**
> We clarify that our framework is not limited to simple mathematical mappings. By integrating multiple tools, FORGE leverages a multi-paradigm computational system capable of modeling diverse domains well beyond pure arithmetic. As documented in its capability scope(https://www.wolfram.com/mathematica/), Mathematica natively supports modeling in **Machine Learning**, **Scientific & Medical Data** , **Geography**, **Engineering**, and even **Social, Cultural & Linguistic Data**. This extensive coverage is complemented by the rigorous logical capabilities of solvers like Z3, which have been proven effective in professional fields outside of STEM. For instance, Bowles et al. (2019) utilized bounded satisfiability checking (the underlying logic of Z3) for "Early Verification of Legal Compliance," proving its applicability to contractual logic, while Feng et al. (2023) developed a framework for "Automated conflict detection and resolution in medical guidelines." Consequently, this combination of general-purpose symbolic computation and logic-based verification ensures that FORGE is methodologically extensible to rule-based tasks across Law, Medicine, and Social Sciences.
>
> We appreciate the reviewer's feedback regarding the documentation. A basic README is already included in the website to guide the installation and initial usage. We are currently further refining the detailed documentation to ensure the package is fully user-friendly for the camera-ready version. The complete code, datasets, and instructions can be accessed at our anonymous website: https://sites.google.com/view/formalbench.

---

### Official Review · Reviewer_t2No · 2025-10-29

**Soundness:** 3
**Presentation:** 2
**Contribution:** 2
**Rating:** 2
**Confidence:** 3

**Summary:**

The paper presents a formal methods-based evaluation framework designed to rigorously probe the systematic generalization abilities of large language models. FORGE addresses the limitations of static benchmarks, such as susceptibility to data contamination and a lack of solution verification, by automatically synthesizing fresh and formal benchmarks from existing datasets. It uses controlled randomization to generate new, structurally equivalent problem instances for every test run.

Experimental results demonstrate a dramatic accuracy drop in top-performing LLMs when tested on FORGE's randomized and verified instances. This persistent performance decline, even after controlling for problem difficulty, highlights critical weaknesses and persistent gaps in LLMs' systematic reasoning and generalization capabilities.

**Strengths:**

1. A complete pipeline for transforming natural language problems into dynamic, verifiable formal artifacts. It uses a three-layered verification method (static, dynamic, and semantic checks) that ensures the correctness and well-posedness of every evaluation instance through formal solvers.
2. A metric that quantifies problem hardness along four dimensions: expression complexity, reasoning complexity, and computational space and time requirements.
3. The proposed stepwise prompting method decomposes complex formal problems into verifiable intermediate reasoning steps. This not only provides powerful diagnostic insights into LLM failures but also offers a structured resource for future step-by-step model training.

**Weaknesses:**

* The proposed framework relies on transforming problems from traditional, existing datasets into a formal, dynamic format. And this transformation process does not guarantee 100% success rate, potentially limiting the diversity and introducing bias in the generated evaluation instances.

* The conclusion of the paper is not surprising and several prior works such as GSM8k-Symbolic have already shown that LLMs struggle when evaluated on perturbed version of existing benchmarks. The novelty of the findings is limited.

* The requirement for formal verification using external solvers introduces additional computational overhead compared to standard, static evaluations. This higher cost could impede large-scale or real-time model evaluation and development iteration.

* The paper presentation has significant room for improvement. For example, Sections 2 and 3 have a lot of overlapping content, reporting R1-R3 is a waste of paper space while the mean and standard deviation would suffice.

**Questions:**

1. Section 2 and Section 3 have a lot of overlapping content. Can the authors clarify the distinction between these two sections and consider merging them for better clarity?

---

> ### Author Response · Authors · 2025-11-25
> **Response to Reviewer t2No**
>
> We thank the reviewer for the detailed feedback and address the main concerns below.
>
>
> ### **1. On the success rate of the formalization pipeline**
>
> We acknowledge that the transformation from natural-language problems to formal, dynamic instances does not achieve a 100% success rate. However, the goal of our work is not to maximize conversion coverage. For those wrong formalized problems, we assume all the LLMs will give incorrect answers because the possibility of catching the correct answer with a wrong problem by accident is quite low. In that case, those incorrect problems can be ignored, since all LLMs will give wrong answers.
>
> Instead, our primary focus is on quantifying the performance *gap* between LLMs. As long as a sufficiently large verified subset exists, a clear and stable gap reliably indicates systematic weaknesses. In practice, the observed robustness gap persists across all models.
>
> ### **2. On novelty relative to prior work**
>
> Prior work indeed shows that LLMs may struggle on symbolic or perturbed versions of existing datasets. We would like to highlight that our paper already includes a detailed comparative analysis of such work, including GSM8k-Symbolic, in the related-work section. Our contribution goes beyond confirming that LLMs degrade on perturbed inputs, as the other reviewers agree. Specifically:
> (1) We present a full pipeline that automatically transforms natural-language tasks into formally verified dynamic artifacts, supported by static, dynamic, and semantic checks.
> (2) We introduce a four-dimensional hardness metric (expression complexity, reasoning complexity, computational time, and computational space) and analyze how performance changes along these axes.
> (3) We propose a stepwise prompting method that decomposes problems into verifiable intermediate reasoning steps, providing interpretable diagnostic insights. We will further highlight these distinctions in the introduction and related-work sections.
>
>
> ### **3. On the computational overhead of external solvers**
>
> Table 8 shows the cost. While using external solvers adds computational overhead, this overhead is negligible compared to LLM inference. Solver calls take only a tiny fraction of the total evaluation time, meaning that the overall pipeline remains substantially cheaper than running the model itself. Since our benchmark is designed for offline evaluation, the solver overhead does not affect scalability.
>
> ### **4. On Section 2–3 overlap and the reporting of R1–R3**
>
> We appreciate the reviewer’s suggestion regarding presentation clarity. Sections 2 and 3 contain limited overlap because one describes the conceptual framework and the motivating example while the other provides implementation details. Nonetheless, we will reorganize these sections to reduce redundancy and sharpen their separation.
>
> Regarding R1–R3, these quantities are not merely additional statistics. They are tied to our hardness metric in the appendix and capture how difficulty dimensions shift under different randomization. One kind of randomness has its own difficulty. Those information cannot be fully conveyed by reporting only means and standard deviations. However, we agree that the presentation can be streamlined. We will keep a concise summary in the main text and move detailed tables of R1–R3 to the appendix.

---

### Official Review · Reviewer_ZMXE · 2025-10-30

**Soundness:** 2
**Presentation:** 3
**Contribution:** 3
**Rating:** 6
**Confidence:** 2

**Summary:**

This paper proposes FORGE, a framework for evaluating LLM systematic generalization by formalizing problems into executable code, generating fresh instances through parameter randomization, and verifying correctness through formal methods. The authors test on GSM8K, MATH, Physics, and Chemistry datasets, reporting significant accuracy drops (20-40%) for SOTA models on randomized versions.

**Strengths:**

S1. The paper tackles two fundamental problems in LLM evaluation, data contamination and systematic generalization, with a principled approach that could have broad impact on the field.

S2. The verification pipeline is well-designed, combining static compilation checks, dynamic solver execution, and semantic verification via judge LLMs. This multi-layered approach increases confidence in benchmark quality.

S3. The formalization enables principled difficulty metrics based on expression complexity, reasoning steps, and computational complexity (space/time). This is a significant improvement over heuristic difficulty labels in existing benchmarks.

S4. Partial demonstration of the framework beyond mathematics (physics, chemistry) shows broader applicability and addresses a limitation of prior work that focuses exclusively on mathematical reasoning.

S5. Consistent performance drops across 7 diverse SOTA models provide moderately compelling evidence.

**Weaknesses:**

W1. Formalization coverage and limitations are somewhat unclear:

-  Are there any theoretical results on what kinds of problems are formalizable with your approach ?
- Success rates vary widely: 81.54-97.62% (Table 5) (presumably the less 'formalized' the domain the worse it gets)
- No analysis of systematic biases in what can/cannot be formalized
- Limited to 3 formal systems, many reasoning types may be excluded
- Generalizability to other domains  questionable

W2. Methods section needs clearer explanation of how problems are perturbed

- It is clear enough to imagine how problem variables are perturbed, but better explanation on how structure (depth, complexity etc.) is modified (if at all) is needed

W3. Needs better justification for their claim that LLMs exhibit poor systematic generalization. Although this is probably true, the logical step between lower accuracy on perturbed problems -> poor systematic generalization should be made more explicit by clearly defining systematic generalization in the context of the problems they evaluate

W4. Needs better analysis demonstrating clearly what aspects of systematic knowledge recomposition are failing in LLMs for the evaluation part of the contributions to be meaningful

- The authors should provide a better taxonomy of the various problem attributes that are being perturbed in the generated benchmarks in order to gain insights into failure modes that are better than "LLMs fail on novel problems".

**Questions:**

Q1. Could the authors provide more information about how the hyperparameters α, β, γ, δ were selected in difficulty quantification?

Q2. a) It is unclear how general this approach can be if it relies on problems being formalizable through a relatively simple mapping. For example, most mathematical problems of interest (e.g. IMO-level and beyond) seem to be hard to formalize as a computer program. b) As I also mentioned in weaknesses, could this approach generalize to scientific domains with less rigorous formalization and self-containment (e.g. involving pragmatics and tacit knowledge within a reasoning problem)? Most reasoning problems (even in pure mathematics) tend to lack the self-containment aspect found in GSM8K-style problems.

Q3. Could the authors provide clarification on whether problem complexity is perturbed on a structural level, and if so, in what ways (branching factor, recursion depth, length, maximum stack size etc.)? Did you perform any analysis on disentangling the effect of structural differences from variable perturbations?

Q4: Could the authors provide more details on what kinds of systematic issues were observed in solutions for datasets with degraded performance?

Q5. Could the authors give a more explicit operational definition of what they mean by systematic generalization, and how their experiments demonstrate a lack of it in LLMs? A short paragraph giving some definitions in the main text could help.

---

> ### Author Response · Authors · 2025-11-25
> **Response to Reviewer ZMXE**
>
> ### **1. About Generalizability**
> Theoretically, Z3 and cvc5 cover the fragment of First-Order Logic combined with background theories, including Linear Integer/Real Arithmetic and Non-linear Real Arithmetic. This scope theoretically guarantees decidability for the vast majority of algebraic and arithmetic reasoning tasks found in benchmarks like GSM8K and MATH. For domains requiring symbolic calculus or complex scientific computing (beyond SMT scope), our framework integrates Mathematica, ensuring broad coverage from discrete logic to continuous scientific modeling.
>
> ### **2. Numeric Perturbation as Structural Change**
> We agree that FORGE primarily evaluates template-level generalization via parametric randomization. However, we clarify that in scientific domains, numeric substitution often entails fundamental structural transformation, rather than mere value recalculation.
>
> **Case Study:** Consider the oxidation of alcohols using acidic $KMnO_4$. Our framework parameterizes the **position locant** (the numeric index indicating the functional group's location) as variable $N$.
>
> * **Seed Problem ($N=1$):** "What is the major product when **1-butanol** is oxidized?"
>     * *Structure:* The index $1$ defines a **Primary Alcohol** ($-CH_2OH$).
>     * *Reasoning:* Primary alcohols oxidize to aldehydes and then fully to **Carboxylic Acids**.
>     * *Answer:* Butanoic Acid.
>
> * **Randomized Variant ($N=2$):** "What is the major product when **2-butanol** is oxidized?"
>     * *Structure:* The index $2$ shifts the functional group, creating a **Secondary Alcohol** ($-CH(OH)-$).
>     * *Reasoning:* Secondary alcohols oxidize to **Ketones** and cannot oxidize further (due to the lack of $\alpha$-hydrogen).
>     * *Answer:* Butanone.
> While the perturbation appears to be a simple numeric substitution ($1 \to 2$), it triggers a **categorical shift** in the chemical species (Primary $\to$ Secondary). This forces the model to traverse a completely different reasoning branch (Acid formation vs. Ketone formation), validating that FORGE evaluates structural understanding through parametric variations.
>
> ### **3. On Systematic Bias in Formalization**
>
> We acknowledge the reviewer's concern regarding potential systematic bias in what can or cannot be formalized. As in our discussion of the formalization success rate, we acknowledge that our pipeline acts as a filter, prioritizing problems with clear logical structures. However, the goal of our work is not to achieve universal coverage of all reasoning types, but to rigorously quantify the performance gap within a verifiable subset.
>
> Crucially, any "systematic bias" introduced by the selection of formalizable problems affects both the **Original Set ($S_{orig}$)** and the **Randomized Set ($S_{rand}$)** equally, as they share the exact same underlying formal structure.
> The fact that LLMs achieve high accuracy on $S_{orig}$ (e.g., ~98% on GSM8K) demonstrates that the "selected" or "formalized" problem types are well within the models' capabilities.
>
> Since the structural bias is constant across $S_{orig}$ and $S_{rand}$, the significant performance drop observed on $S_{rand}$ cannot be attributed to the nature of the problems being "unsuitable" or "biased." Instead, this stable gap reliably isolates the failure of systematic generalization. Specifically, the inability to adapt to parameter shifts within a structure the model has already proven it can solve.

---

### Official Review · Reviewer_1RLo · 2025-11-01

**Soundness:** 2
**Presentation:** 4
**Contribution:** 2
**Rating:** 4
**Confidence:** 4

**Summary:**

The authors present FORGE, a framework for evaluating LLM reasoning. The work is motivated by the problem of "data contamination," where LLMs may have memorized answers to static benchmarks like GSM8K. FORGE's method is to auto-formalize problems from these benchmarks into executable code for solvers (like Z3). It then generates novel, unseen problems by randomizing parameters (e.g., numbers) within these formal "templates." These new problems are verified for correctness by the solvers. The authors find that LLMs, which score highly on the original benchmarks, show a large drop in accuracy on these new, randomized versions.

**Strengths:**

Solver-Based Ground Truth: The paper's best idea is using formal solvers (Z3, cv5c) for verification. This "Tri-Verify" process creates a rock-solid, unambiguous ground truth for every problem, which is a major step up from benchmarks that might have errors or ambiguous answers.

Good Diagnostics: The "Stepwise Prompting" module (Table 1) is a smart way to diagnose why a model is failing. By breaking the problem into logical sub-questions, it can pinpoint the exact step where the model's reasoning breaks down, which is far more useful than a simple pass/fail grade.

Solid Experimental Controls: Using a "Difficulty Quantification" metric to prove their new problems aren't just "harder" was the right move. It effectively isolates the variable, showing the accuracy drop is due to the model's lack of generalization, not a spike in problem difficulty.

**Weaknesses:**

A Limited Definition of Generalization: The paper's central weakness is its narrow definition of "generalization." FORGE only tests "template-level" generalization which is whether a model can apply the same rule to new numbers. It doesn't test true "systematic generalization," which would involve combining multiple known rules in novel ways (i.e., compositionality). The "rules" themselves are never changed or combined, only the parameters.

Claims as Training Tool Lack Experimental Support: The authors rightly suggest FORGE could be a "potential training resource" by generating step-by-step verified data. However, this claim is presented without any experimental support. The paper lacks a crucial experiment: fine-tuning a model on FORGE-generated data and then testing it on a new, held-out set of instances from the same templates. This makes the "training" aspect an unsubstantiated claim, when it might be the framework's most valuable contribution.

Needs Discussion of Parallel Work: The authors should situate their work relative to efforts like "Certifying Knowledge Comprehension in LLMs" (CKC, arXiv:2402.15929). Both frameworks generate novel instances from a formal specification (FORGE uses solver code; CKC uses knowledge graphs) to avoid contamination. However, their goals and methods differ: CKC tests knowledge comprehension using rich semantic noise (distractors, shuffling), which is a stronger test of robustness in that domain. FORGE tests mathematical reasoning using parametric noise. A discussion would clarify that FORGE's unique strengths are its solver-verified ground truth, TriVerify, and its diagnostic "Stepwise Prompting," which are distinct from CKC's goals and not confuse the reader about the primary contribution being a template based way of evaluating models which CKC already does.

**Questions:**

tGiven the authors' suggestion that FORGE is a good training resource, would they consider adding experiments to validate this? For example, an experiment showing how model performance improves after being fine-tuned on FORGE-generated synthetic data? This would help demonstrate its value for improving model robustness on specific problem classes.

Following that, how do the authors distinguish between this "template mastery" (which can be trained) and the deeper, "compositional" generalization (applying rules to new problem structures)? The paper defines generalization as applying a rule to new parameters. Should "systematic generalization" require more, such as composing multiple rules or applying rules to new problem structures? I am curious how the authors see FORGE addressing this deeper, compositional aspect of generalization, which goes beyond mastering a single, fixed template

---

> ### Author Response · Authors · 2025-11-25
> **Response to Reviewer 1RLo**
>
> ### **1.Scope of Generalization**
>
> We agree that the primary type of generalization evaluated by FORGE is template-level generalization. However, FORGE is not limited to numeric parameter substitution. Some of our tasks involve generating structurally different problem variants by altering the underlying functional groups of an organic compound, rather than merely changing numeric values.
>
> For example, in some of our chemistry problems, replacing the functional group (e.g., alcohol → aldehyde → ketone) changes the space of structurally valid molecules and the applicable reaction or classification rules. These transformations therefore generate new task variants that are not reducible to the same template with different parameters—they involve genuine changes in the compositional structure of the underlying rules.
>
> We did not emphasize this aspect in the main experiments because such tasks naturally carry greater intrinsic difficulty. To ensure fair comparisons across models, we kept the difficulty parameters constant and focused on the controlled setting. Nevertheless, we appreciate the reviewer’s suggestion, and we will expand this part and add more explicit structure-level and compositional variations to better address deeper forms of generalization.
>
> ### **2.Experiment on Training Resources:**
>
> We agree with the reviewer that our claim that FORGE could serve as a potential training resource currently lacks full experimental validation. The purpose of this paper is to introduce a benchmark rather than large-scale fine-tuning experiments require substantial computational resources beyond the scope of this submission. We are actively conducting small-scale exploratory fine-tuning experiments using FORGE-generated data to preliminarily assess its potential benefits. We will report initial observations in the appendix and clearly position these findings as ongoing work.
>
> ### **3.About Parallel Work**
> We appreciate the reviewer’s suggestion to Certifying Knowledge Comprehension in LLMs. We have now added a clear comparison in the revised version.
>
> Importantly, CKC targets a fundamentally different problem setting. CKC focuses on knowledge comprehension over knowledge graphs, using graph-structured context to generate large distributions of multi-hop factual queries. Its goal is to certify whether LLMs can retrieve correct factual entities from noisy, graph-based prompts.
>
> This is fundamentally different from our setting.  Our framework evaluates **mathematical reasoning under formal verification**, where each test instance is converted into executable formal code (Z3/Mathematica) and validated through rigorous static, dynamic, and semantic checks. Thus, the two frameworks are complementary but non-overlapping, and comparing them clarifies rather than diminishes the novelty of our contribution.

---

### Comment · Area_Chair_WP4r · 2025-11-28
**Respond to the authors’ rebuttal**

Dear Reviewers,

As the discussion phase is nearing its end, please read and respond to the authors’ rebuttal, particularly if it addresses your concerns. Your response is valuable for the final decision.

Reviewer ZMXE, you currently hold the most positive rating. Have the authors’ response or other reviewers' comments affected your opinion?

---

### Note · Program_Chairs · 2026-01-17
**Submission Desk Rejected by Program Chairs**

The following references in this submission do not refer to real documents and/or have major errors in bibliographic information:

 Yasaman Razeghi, Adam Roberts, Katherine Lee, and Colin Raffel. The impact of memorization on the effectiveness of retrieval-augmented language models. arXiv preprint arXiv:2202.07646, 2022.